# Effectiveness of an Inpatient Virtual Reality-Based Pulmonary Rehabilitation Program among COVID-19 Patients on Symptoms of Anxiety, Depression and Quality of Life: Preliminary Results from a Randomized Controlled Trial

**DOI:** 10.3390/ijerph192416980

**Published:** 2022-12-17

**Authors:** Sebastian Rutkowski, Katarzyna Bogacz, Oliver Czech, Anna Rutkowska, Jan Szczegielniak

**Affiliations:** 1Faculty of Physical Education and Physiotherapy, Opole University of Technology, 45-758 Opole, Poland; 2Specialist Hospital of the Ministry of the Interior and Administration in Głuchołazy, 48-340 Głuchołazy, Poland; 3Department of Physiotherapy, Wroclaw University of Health and Sport Sciences, 51-612 Wroclaw, Poland

**Keywords:** pulmonary rehabilitation, virtual reality, COVID-19, anxiety, stress

## Abstract

Forms of rehabilitation for patients after COVID-19 are gaining interest. The purpose of this study was to investigate and compare an innovative in-hospital pulmonary rehabilitation programs augmented with training elements performed in virtual reality. This randomized controlled study included 32 patients enrolled in post-COVID-19 rehabilitation at a Public Hospital in Poland. The rehabilitation models included exercise capacity training on a cycle ergometer, breathing and general fitness workout, resistance training, and relaxation. The forms of training and relaxation differed between the groups: the experimental group employed virtual reality, and the control group used a traditional form of therapy. Exercise tolerance was assessed using a 6 min walk test (6 MWT), while psychological parameters were evaluated using the Hospital Anxiety and Depression Scale (HADS) and the brief World Health Organization Quality of Life Scale (WHOQOL-BRIEF). The analysis of the post-rehabilitation results showed a statistically significant improvement in both groups regarding depression (VR: 6.9 (3.9) vs. 4.7 (3.5), *p* = 0.008; CG: 7.64 (4.5) vs. 6.6 (4.8), *p* = 0.017) and anxiety (VR: 8.6 (4.6) vs. 5.6 (3.3), *p* < 0.001; CG: 9.57 (6.0) vs. 8 (4.8), *p* = 0.003). No statistically significant improvements in quality of life were noted in both groups. Moreover, the analysis showed a statistically significant improvement in the exercise capacity in both groups after completion of the rehabilitation program, expressed as a distance in the 6 MWT, as well as a statistically significant improvement in dyspnea in the VR group. To conclude, the analysis of the preliminary data revealed that a 3-week hospital-based pulmonary rehabilitation program for COVID-19 patients led to an improvement in exercise tolerance as well as a reduction in the symptoms of anxiety and depression. The virtual reality-based form of training delivery, despite its attractiveness, did not significantly affect patients’ performance.

## 1. Introduction

Over recent months, SARS-CoV-2 has been confirmed in millions of people around the world. The virus spreads through the respiratory tract (especially drops arising from coughs, sneezes and talking) and through contaminated surfaces and biological substances [1]. Clinical symptoms in patients, depending on the variant of the virus, include fever, sore throat, cough, fatigue, or gastrointestinal or neurological symptoms. Respiratory failure also occurs, as well as heart and kidney damage [2]. Initial studies indicated that approximately 60 days after the first COVID-19 symptom onset, only one out of ten patients previously hospitalized for SARS-CoV-2 were reported to be essentially free of any infection-related symptoms, while one-third had one or two symptoms, and over a half had three or more symptoms [3]. Furthermore, the COVID-19 epidemic has given rise to new emotional stressors, e.g., social isolation, physical distancing, employment loss and uncertainties about the future. Psychological problems in different age groups have occurred with different intensities and duration during the pandemic. In the initial stage of the pandemic, seniors were the most psychologically affected by the direct threat to their health and life, as well as people who were suddenly deprived of their resources due to the lockdowns. In the following months, the population began to experience long-term effects in the form of a safety hazard for those whose professional work seemed to be safe before. Long-term, chronic stress also affected chronically ill people. It is known that these people have a higher risk of severe COVID-19 compared to the general population, even resulting in death; in addition, other serious difficulties, such as limited access to medical services, lack of beds in hospitals, lack of medical staff have caused a significant deterioration in the treatment of their basic diseases. Increasing stress, anxiety and even depression worsen the somatic conditions of the elderly and of those suffering from other chronic diseases and can be a cause of premature death, even in the absence of a COVID-19 infection. During the previous SARS outbreak, patients reported fear, loneliness, fatigue and anger. Studies also indicated that patients felt anxiety as a result of fever and insomnia [4]. Zheng reported a high prevalence of psychiatric disorders among survivors of COVID-19, increased depressive states and post-traumatic stress disorder [5]. 

A large number of symptoms urge patients to seek specialist medical support. The side effects of the infection have a significant negative impact on the general health of patients and often make a prompt hospitalization necessary. At the beginning of the pandemic, the health services in most parts of the world were not prepared for the intake of patients to hospital wards. Two years after, the number of facilities dedicated to rehabilitating COVID-19 patients and COVID-19 survivors is slowly increasing. However, it is still a small percentage of the medical service in many countries. Comprehensive rehabilitation approach programs involving a multidisciplinary team offering medical, physiotherapeutic and psychological interventions are expected to be offered to patients with post-COVID-19 symptoms [6]. The available rehabilitation programs for SARS-CoV-2 survivors are mostly based on the pulmonary/cardiac rehabilitation model, where the main component includes capacity training aimed at improving exercise tolerance and lung function, likewise reducing the level of dyspnea and fatigue [7]. However, experts believe such programs should be augmented with psychological support to improve the quality of life (QoL) and reduce symptoms of depression or anxiety. The experts from the European Respiratory Society (ERS) and the American Thoracic Society (ATS) indicate that a rehabilitation program should begin between 6 and 8 weeks after hospital discharge [8]. Recommendations with regard to the choice of the form of treatment are open. It can be noted that the medical staff still prefers the use of telerehabilitation, remote consultations and self-treatment programs for the patients. Unfortunately, unsupervised exercise, pharmacotherapy or treatment, in general, carry a high risk for patients in terms of their recovery process. Inpatient rehabilitation is much safer and more effective because the patient is constantly corrected and supervised. Therefore, all precautions should be taken to enable the in-patient rehabilitation of COVID-19 patients. Another argument in favor of inpatient rehabilitation is the unpredictable dynamics of changes in the pandemic process. The remaining duration of the pandemic is difficult to predict, and the long-term effects of the disease are not known. After epidemiological observation, it is already clear that COVID-19 appears in waves; hence, it seems unlikely that this virus will be completely defeated in the coming years. Therefore, the use of substitution measures based on remote treatment may have unexpected consequences for the patients. This study presents a stationary rehabilitation program enriched with modern, unconventional methods of treatment. 

The aim of this study was to compare the effectiveness of two forms of hospital pulmonary rehabilitation among COVID-19 patients on exercise tolerance and mental well-being indices. We assumed the following hypotheses: (1) by the completion of the 3-week inpatient rehabilitation program, the patients will show reduced symptoms of anxiety and depression, (2) the patients participating in a virtual-reality relaxation program will achieve greater improvement, (3) the 3-week pulmonary rehabilitation program will improve quality of life among the COVID-19 patients.

## 2. Materials and Methods

### 2.1. Participants

This study enrolled 32 patients who participated in inpatient pulmonary rehabilitation, previously affected by SARS-CoV-2. Patients meeting the criteria of inclusion were randomly divided to one of two groups: VR group (VR) or control group (CG). Randomization was performed using Research Randomizer (1:1 ratio), a web-based service that offers random assignment. The main characteristics of the patients are presented in Table 1. Inclusion criteria included: women and men aged 40–80 years, diagnosed with coronavirus disease-19. The exclusion criteria were: no consent to participate, as well as active pneumonia diagnosed by X-ray or documented heart disease (stable or unstable), status after CABG, PTCA, uncontrolled hypertension, insulin-dependent diabetes mellitus, inability to exercise independently or musculoskeletal/neurological conditions that would prevent the completion of the course, lung cancer, cognitive impairment or Mini-Mental State Examination <24. This study implemented a randomized control trial study design, approved by the Bioethical commission Opole Medical Chamber in Opole (Approval Number: No. 343, 25 November 2021), registered in ClinicalTrials.gov (NCT05244135) and carried out in accordance with the Declaration of Helsinki guidelines [9]. Figure 1 presents the study flow (Figure 1).

### 2.2. Interventions

An intensive in-patient 3-week rehabilitation program, five times a week, was used as the intervention treatment. Short-term programs have been found to exhibit clinically important improvements in exercise capacity, dyspnea, quality of life and lung function in patients with either COPD [10,11] or lung cancer [12]. The authors’ pulmonary rehabilitation program was programmed based on previous experience in patients with COPD [13]. A comprehensive pulmonary rehabilitation program for COVID-19 patients with combined treatment focused on exercise capacity increase, lung function restoration and mental health support, developed by a multidisciplinary team was introduced [14]. Based on the patient’s submaximal exercise tolerance test results, qualification for one of the respiratory physiotherapy models differing in therapy intensity was determined. The rehabilitation models included exercise capacity training on a cycle ergometer, breathing exercises, general fitness and resistance training, as well as relaxation. The training heart rate limits on a cycle ergometer vary by model (model A, 80% of the submaximal heart rate, model B, 70%, model C, 60%); in model D, the heart rate increases during the exercises by 20–30% in relation to the heart rate at rest. Every procedure was performed once a day. A detailed explanation of the rehabilitation program was previously presented [14].

#### 2.2.1. Relaxation in Virtual Reality

A VR TierOne device (Stolgraf^®^, Stanowice, Poland) was used as the VR source. The workstation consisted of HMD (head-mounted display) goggles connected to a computer and a chair on which relaxation was conducted (Figure 2). The software was developed with the aim of providing calmness and mood improvement as well as motivation and cognitive activation to the patients. The software is based on the Ericksonian psychotherapy approach and presents a virtual therapeutic garden. The garden is a metaphor for the patient’s wellbeing: at the beginning, it seems untidy and grey then, gradually, becomes more colorful and alive, which symbolizes the process of recovery of energy and vigor [15]. Previous studies have shown that the device, when incorporated into the rehabilitation process, improves the symptoms of depression, anxiety and stress in pulmonary and cardiac patients [16,17].

#### 2.2.2. Exercise Training in VR

The form of exercise training differed between the groups. In the control group, training was conducted in the traditional form, on a cycle ergometer without additional audio-visual stimuli. The VR group conducted training on a bicycle ergometer with an HMD using “Virtual Park” software developed by STIIMA-CNR. The software brought the participants to a sunny island where they conducted a bike ride enhanced with realistic elements and sound effects simulating real-life situations. The training station consisted of a COSMED bicycle ergometer, VR goggles and physiological sensors—an HR band and a pulse oximeter (Figure 3). The system was designed for COPD patients, with subsequent adaptation to the research project [18].

### 2.3. Measurement

At baseline, the participants completed a self-administered sociodemographic questionnaire. The questions included gender, marital status, education, likewise history of hospitalization for SARS-CoV-2 infection. The collected data were used for group characterization and statistical analysis.

#### 2.3.1. Hospital Anxiety and Depression Scale

The Polish translation of the Hospital Anxiety and Depression Scale (HADS) was incorporated to examine depression and anxiety. The questionnaire is validated as a reliable method of assessing anxiety and depression [19]. It consists of 14 questions scored on a 4-point (0 to 3) scale. Two separate subscales are included in the questionnaire: the HADS-A for anxiety and the HADS-D for depression. A score of 8 out of 21 on each subscale is considered a cut-off point, meaning that the patients are classified into a group characterized by anxiety or depression disorders. The HADS-A had optimal cut off ≥8 (sensitivity 0.89, specificity 0.75) [20]. The Cronbach’s alpha for HADS-A was α = 0.83 and for HADS-D, it was α = 0.82 [21].

#### 2.3.2. WHO Quality of Life-BREF

Quality of life was examined with a short version of the World Health Organization Quality of Life questionnaire (WHOQOL-BREF). The questionnaire evaluates quality of life in four categories: physical health, psychological health, social relationships and environment. It consists of 26 questions with answers organized on a five-point Likert scale. The WHOQOL-BREF is a cross-culturally valid assessment of quality of life, with an internal consistency alpha ranging from 0.68 to 0.82 in each domain [22].

#### 2.3.3. Exercise Capacity

The exercise capacity was assessed by a 6 min walk test (6 MWT). The 6 MWT test is a valid and reliable measure of exercise capacity in individuals with cardiopulmonary impairment [23]. The test is conducted over a length of 30 m while measuring the distance the patient is able to walk in 6 min. The patients were instructed to walk as far as possible and allowed to move independently and rest as needed while pacing back and forth along a marked walkway. The study was conducted according to the European Respiratory Society and American Thoracic Society guidelines.

### 2.4. Statistical Analysis

All analyses were performed using Statistica 13 software (StatSoft, Cracow, Poland) and JASP software (JASP Team, Amsterdam, Netherlands) [24]. Prior to the analysis, the data distribution was tested using the Shapiro–Wilk test for normality. The statistical significance level was set at α = 0.05. Categorical variables are presented as numeric values and percentages. Continuous variables are presented as mean ± standard deviation (SD) or median and interquartile range [IQR]. We used G*power 3.1.9 software (Universität Düsseldorf, Düsseldorf, Germany) for sample size calculation. The calculation was based on the F test and repeated-measures within = between factors: the type I error rate was set at 5% (alpha-level 0.05), the effect size of the main outcomes was 0.234, and the type II error rate produced 85% power. The sample size was calculated based on previous studies on the effectiveness of immersive VR relaxation [25], and it was determined that 30 patients should be enrolled. The differences in pre-post rehabilitation values were analyzed by the paired t-Student’s test or the Wilcoxon test, depending on the distribution of the variables. Multiple linear regression (stepwise) was used to identify the association between headache during COVID-19 and depression and anxiety symptoms and stress levels. Prior to the multiple linear regression analysis, the assumption of a linear relationship (using the point biserial correlation coefficient) between the outcome variable and the independent variables was tested. The effect sizes were calculated with Cohen’s d. An effect size ≥0.20 was considered small, while an effect size ≥0.50 was considered medium, and an effect size ≥0.80 was considered large [26].

## 3. Results

### 3.1. Characteristics of the Group

This study enrolled 32 patients who participated in inpatient pulmonary rehabilitation at the Specialist Hospital in Glucholazy. Among the patients, 6 (18.75%) had primary or vocational education, 12 (37.5%) had higher education, and 14 (43.75%) had secondary education. Twenty-one patients (65.25%) were professionally active, and 11 (34.75%) were retired; 27 (84.37%) of the study participants were married or in an informal relationship, and 7 (15.63%) were widowed, divorced or single. Twenty-one patients (65.25%) were hospitalized due to COVID-19, while 11 (34.37%) stayed at home during the illness. There were no statistically significant differences between groups with specific sociodemographic characteristics.

### 3.2. Assessment of Depression and Anxiety

The analysis of the results showed that the mean score in the experimental group for depression (HADS-D) was 7.2 (±3.7) points. In the control group, 7 patients (50%) exceeded the cut-off point (score ≥ 8 points) and were prone to depressive disorders, whereas in the VR group, 8 patients (44.4%) exceeded it. The mean score for anxiety (HADS-A) was 9.0 (±4.4) points, and 7 patients (50%) in the control group and 11 patients (61.1%) in the VR group, exceeding the cut-off point (score ≥8 points), were prone to anxiety (Table 2). The analysis of the post-rehabilitation results showed a statistically significant improvement in both groups (Figure 4). The number of subjects presenting symptoms of depression decreased to 4 in the control group and to 3 in the VR group, with small and medium effect size.

### 3.3. Assessment of Quality of Life

The analysis of the quality of life scores showed no significant changes in both groups (Table 3). Within the control group, deterioration of quality of life in the environmental domain was noted (Figure 5). 

### 3.4. Assessment of Functional Capacity

The analysis showed a statistically significant improvement in the exercise capacity in both groups after completion of the rehabilitation program, expressed as a distance in the 6 MWT. The achieved average improvement of 56.9 m in the VR group and 39.2 m in the control group was found to be beyond the minimum detectable change for this test (Table 4). Moreover, a statistically significant decrease in the dyspnea levels in the VR group was noted (Figure 6).

### 3.5. Predictors

Stepwise multiple regression was used to examine how baseline outcomes variables related to functional capacity, quality of life and depression and anxiety could explain a statistically significant amount of variance in HADS Total (Table 5). For change Delta HADS Total (difference between the final value and the initial value), baseline Fatigue and baseline HADS A were found to be significant predictor variables, accounting for 45% of the variance in these models (*p* < 0.001).

## 4. Discussion

### 4.1. Results Interpretation

Concerning our primary outcome, the results showed that a 3-week inpatient rehabilitation program led to the improvement of patients’ mental health and functional capacity, without effect on the quality of life. The analysis showed that anxiety and depression symptoms are frequent problems in patients during post-COVID rehabilitation; in fact, 56.25% of the analyzed cohort was prone to anxiety, while 46.87% of the patients were prone to depressive disorders. However, the data analysis revealed a significant reduction in the symptoms of anxiety and depression in both groups. Regarding the functional capacity of the patients, the analysis of the results showed a statistically significant improvement in exercise tolerance. No changes in quality of life were noted, probably due to the program being too short. Therefore, the results confirmed the first hypothesis. Regarding the second hypothesis, no superior effect of the virtual reality-based intervention was found. Similarly, the third hypothesis could not be confirmed. The results obtained suggest that the modern therapeutic form did not significantly affect the effectiveness of the inpatient rehabilitation program for individuals after COVID-19. Noteworthy findings presented in this report are preliminary findings, and the results will be re-evaluated once the project is completed on a larger research sample. However, it is important to compare the results we obtained with the other results of the study on the group of patients after COVID-19.

### 4.2. Discussion of Related Research

It is important to refer to current reports in the analyzed area; however, the number of publications is still limited. A study by Liu et al. [27] investigated the effectivity of 6-week respiratory rehabilitation training on the respiratory function, QoL, mobility and cognitive function in elderly patients with SARS-CoV-2. Functional parameters such as FEV1(L), FVC(L), FEV1/FVC%, DLCO% and the 6 min walk significantly improved in the intervention group. It is worth noting that the incidence of COVID-19 contributed to the deterioration of these parameters in each of the study participants. The authors also found a positive impact of the rehabilitation process on the patients’ quality of life and anxiety levels. Unfortunately, conventional rehabilitation did not significantly affect the symptoms of depression in the studied population, which can impair recovery time and effectiveness. A recent systematic review and meta-analysis by Chen H et al. [7] investigated the effect of pulmonary rehabilitation on lung impairment in patients after COVID-19. It is noteworthy that only three RCTs were included in the analysis. The analysis revealed that pulmonary rehabilitation could improve the exercise capacity measured by the 6-MWT among patients with mild-to-moderate lung impairment associated with COVID-19. However, the authors concluded that the effects on lung function, dyspnea, and quality of life should be interpreted with caution due to inadequate and conflicting data reported across the studies. More rigorous and long-term evidence of the effect of PR among patients, especially those with severe lung impairment, are needed to guide the PR practice for the increasing number of survivors of COVID-19. This highlights a lack of evidence to determine the effectiveness of pulmonary rehabilitation programs for patients after SARS-CoV-2 infection. 

The available literature identifies a relationship between quality of life, anxiety or depressive symptoms and the effectiveness of rehabilitation programs among patients with pulmonary disfunctions. A patient’s well-being also directly influences the rehabilitation process. Therefore, comprehensive rehabilitation of post-COVID-19 patients or patients during the infection should both include the assumptions of conventional rehabilitation and address the psychological needs of these patients. A study by Panagioti et al. showed a reduction of the depression levels made pulmonary rehabilitation more effective and improved the patients’ motivation to cooperate actively in the rehabilitation process [28]. However, anxiety, stress and fear do not concern only pulmonary wards. Abasıyanık et al. [29] investigated the physical activity behavior in people with neurological diseases during the COVID-19 pandemic. According to the authors, the coronavirus pandemic has had a negative impact on the physical activity levels of people with neurological diseases, and this change was related to the worsening of the disease symptoms and to psychosocial factors. Comorbidities are certainly a factor that worsens the condition of patients treated for COVID-19. However, according to Argüder et al. [30], hospitalization for SARS-CoV-2 alone also has serious consequences for the patients’ mental health. The findings showed that the anxiety and depression levels increased in patients hospitalized for coronavirus infection. The epidemic is expected to continue in the future. For this reason, it is thought that it is important to plan interventions on patients to reduce their anxiety and depression levels and to create the necessary support programs. A cohort study by Huang C et al. found, at 6 months after acute infection, that COVID-19 survivors were mainly troubled with fatigue or muscle weakness, sleep difficulties, and anxiety or depression [31]. Patients who were more severely ill during their hospital stay had more severely impaired pulmonary diffusion capacities and abnormal chest imaging manifestations and were the main target population for the intervention of long-term recovery. These findings are in line with the study of Xiong O. et al. [32]. The reports further emphasize that COVID-19 infections cannot be underestimated. The effects of the infection, apart from being burdensome and deteriorating the quality of life, also tend to recur or persist for a long time. The benefits resulting from, e.g., physical activity, which is an inseparable element of pulmonary rehabilitation and VR therapy, seem to be an appropriate measure to prevent the long-term effects of the infection. In addition, therapy using VR has a beneficial effect on the motivation of the patients. The patients are more likely to use an interactive, new therapy than similar interventions involving passive participation.

The pandemic has changed the world, and the health service is expected to adapt to it. Thanks to modern technologies that were once used sporadically in medical facilities, it is possible to improve the effectiveness of the treatments, increase the safety of patients and staff, and raise the standards of treatment. The pandemic contributed to a significant increase in interest in these technologies, which directly impacts the effectiveness of their operation and the level of technological excellence. Further research is needed to improve the procedures of remote, unattended or autonomous treatments, as well as to systematize their application. Along with the development of technology, the direction of research could be chosen based on the assessment of the effectiveness of complementary therapy, using each of the described tools. Research shows that the best therapeutic effects are usually obtained with combination therapy. Therefore, we should look forward to self-care virtual telerehabilitation.

### 4.3. Limitations

Although this study provides evidence for the effectiveness of inpatient rehabilitation programs, we recognize that some limitations should be considered. First, a follow-up assessment could provide valuable information. Second, the analysis of quality of life might have covered too short a period of time to record changes. However, we would like to emphasize that the results obtained are preliminary, and the project is still in the implementation phase.

## 5. Conclusions

The program of hospital-based 3-week pulmonary rehabilitation for COVID-19 patients led to an improvement in exercise tolerance as well as to a reduction in the symptoms of anxiety and depression. Despite its attractiveness, the virtual reality-based form of training delivery did not significantly affect patients’ performance. Given the high number of patients proceeding to post-COVID-19 rehabilitation, it seems important to include programs with relaxation and psychotherapy components in the treatment of this group of patients.

## Figures and Tables

**Figure 1 ijerph-19-16980-f001:**
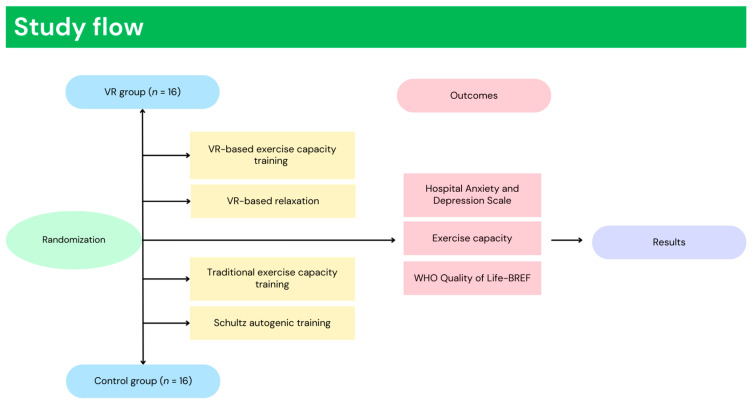
Study flow.

**Figure 2 ijerph-19-16980-f002:**
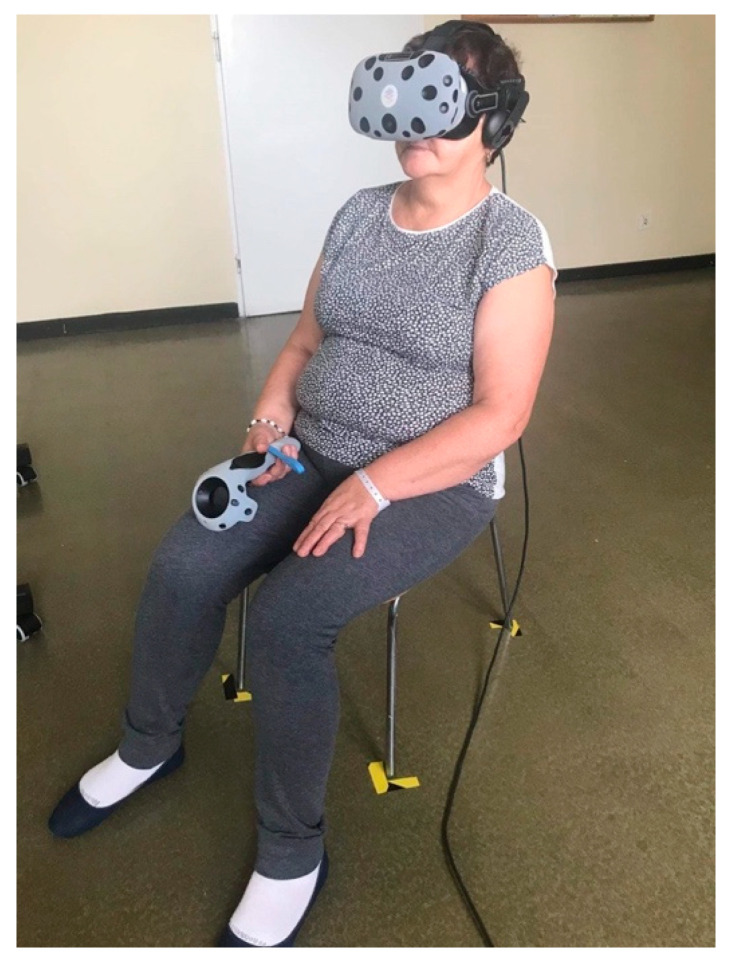
A station for relaxation in VR.

**Figure 3 ijerph-19-16980-f003:**
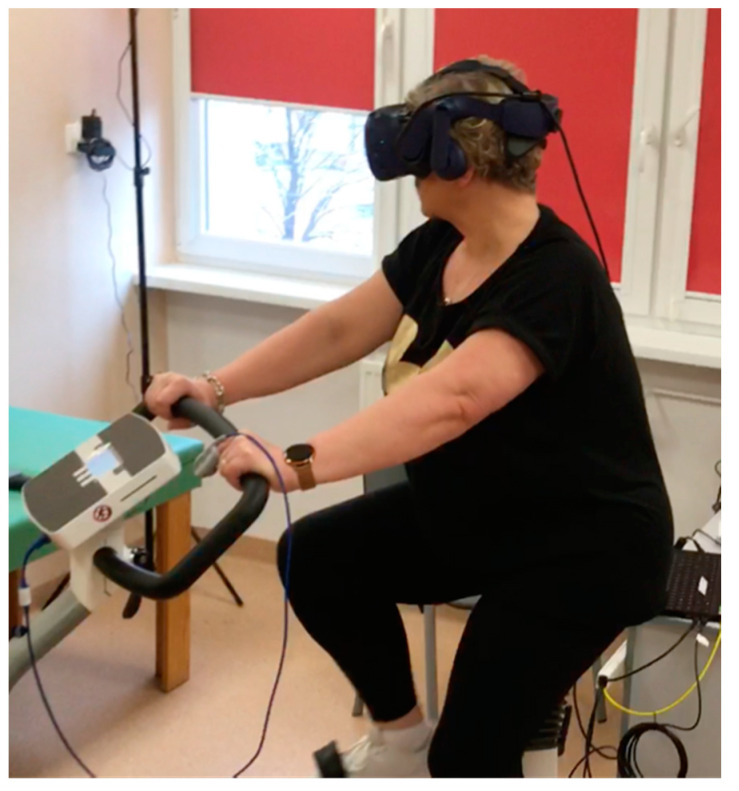
Exercise training in VR.

**Figure 4 ijerph-19-16980-f004:**
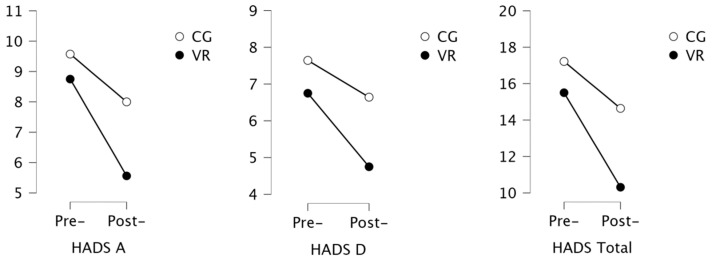
Results of the analysis of anxiety and depression.

**Figure 5 ijerph-19-16980-f005:**
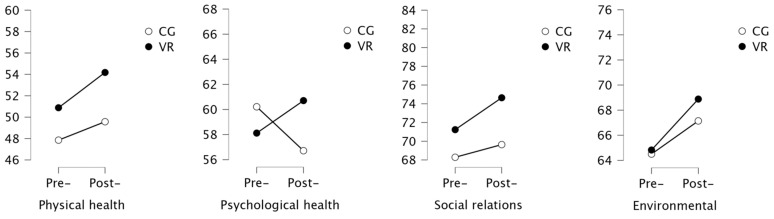
Results of the analysis of Quality of Life.

**Figure 6 ijerph-19-16980-f006:**
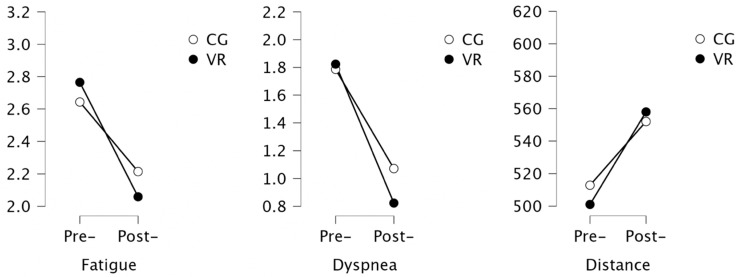
Results of the analysis of exercise capacity.

**Table 1 ijerph-19-16980-t001:** Group characteristics.

Variables	Mean (SD)
Age (years)	57.8 (4.9)
Female, *n* (%)	20 (68.75%)
FVC (L)	3.47 (0.65)
FEV1 (L)	2.76 (0.48)
FEV1% VC	84.6 (4.31)
TLC (%)	5.06 (0.98)

Notes: FEV1: forced expiratory volume for 1 s, FVC: forced vital capacity; FEV1%VC: forced expiratory volume in one second % of vital capacity; TLC: total lung capacity; 6 MWT: 6 min walk test, SD: standard deviation.

**Table 2 ijerph-19-16980-t002:** Analysis of anxiety and depression.

Variable	VR Group			Control Group			
	Pre-	Post-	*p*	Pre-	Post-	*p*	Cohen’s d
HADS-A mean (SD)	8.6 (4.6)	5.6 (3.3)	**<0.001**	9.57 (6.0)	8 (4.8)	**0.003**	0.595
HADS-D mean (SD)	6.9 (3.9)	4.7 (3.5)	**0.008**	7.64 (4.5)	6.6 (4.8)	**0.017**	0.457
HADS Total mean (SD)	15.5 (7.5)	10.3 (6.5)	**<0.001**	17.2 (9.8)	14.6 (8.9)	**<0.05**	0.559

Bold highlights statistical significance, *p* < 0.05. HADS: Hospital Anxiety and Depression Scale; SD: standard deviation.

**Table 3 ijerph-19-16980-t003:** Analysis of the quality of life.

Variable	VR Group			Control Group			
	Pre-	Post-	*p*	Pre-	Post-	*p*	Cohen’s d
Physical health	50.8 (11.3)	51.8 (9.8)	0.618	47.9 (11.8)	52.5 (11.6)	0.154	0.164
Psychological health	59.8 (13)	59.4 (12)	0.906	57.5 (14.3)	58.1 (15.5)	0.966	0.043
Social relationship	71.4 (14.1)	71.8 (9.1)	0.838	67.9 (19.5)	73.1 (17.5)	0.174	0.005
Environmental	64.4 (19)	68.2 (16)	0.168	65.4 (19)	68 (15)	**0.028**	0.064

Bold highlights statistical significance, *p* < 0.05.

**Table 4 ijerph-19-16980-t004:** Analysis of the functional capacity.

Variable	VR Group			Control Group			
	Pre-	Post-	*p*	Pre-	Post-	*p*	Cohen’s d
Fatigue	2.78 (1.06)	2.06 (0.90)	0.055	2.64 (1.08)	2.21 (0.70)	0.095	0.191
Dyspnea	1.72 (1.52)	0.82 (1.13)	**0.033**	1.78 (1.48)	1.07 (1.2)	0.061	0.213
Distance	502 (48.4)	558 (76)	**<0.001**	512 (54.3)	552 (49.1)	**0.006**	−0.091

Bold highlights statistical significance, *p* < 0.05. FEV1: forced expiratory volume for 1 s, FVC: forced vital capacity; FEV1%VC: forced expiratory volume in one second % of vital capacity; TLC: total lung capacity; 6 MWT: 6 min walk test, SD: standard deviation.

**Table 5 ijerph-19-16980-t005:** Baseline functional capacity, quality of life, depression and anxiety as predictors of change in HADS Total.

Variable	*B*	Beta	*t*	*p*	*F*	R^2^
Change in HADS Total				<0.001	11.293	0.455
Baseline HADS A	−1.035	−0.770	−4.737			
Fatigue	−2.885	−0.428	−2.634			

The following covariates were considered but not included: baseline: Dyspnea, Distance, HADS D and WHOQOL-BREF (Physical health, Psychological health, Social relationship, Environmental).

## Data Availability

The data presented in this study are available on request from the corresponding author.

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
