# Peer review of "Effectiveness of an Inpatient Virtual Reality-Based Pulmonary Rehabilitation Program among COVID-19 Patients on Symptoms of Anxiety, Depression and Quality of Life: Preliminary Results from a Randomized Controlled Trial"

_ijerph, 2022, doi:10.3390/ijerph192416980_

Round 1

Reviewer 1 Report

The manuscript of Sebastian Rutkowski et al. gives an overview of preliminary data on the effectiveness of an inpatient virtual reality-based pulmonary rehabilitation Program among COVID-19 patients. First of all, I would like to mention that this is a study with a very small number of patients. Since significant results were nevertheless shown, the study has a certain scientific value. 

(1) In my view, the introduction is far too superficial and should provide a more scientifically correct insight into the subject.

(2) For a better understanding of the material and methods, an overview illustration must be created from my point of view. 

(3) The results are presented much too briefly, furthermore only very few results are listed. If a publication should take place, these must be presented much more extensively and comprehensively. 

(4) Unfortunately, the discussion is absolutely superficial - a comprehensive revision is needed here.

In summary, this is a currently still completely inadequately presented scientific analysis. Since the topic is very interesting from my point of view, I recommend the authors to revise it thoroughly. I am already looking forward to seeing these. 

Author Response

Dear Reviewer,

Thank you for your deep and knowledgeable revision of this manuscript. The following are our answers:

  • In my view, the introduction is far too superficial and should provide a more scientifically correct insight into the subject.
Thank you for pointing this out. The introduction has been improved and this section seems more consistent now.
  • For a better understanding of the material and methods, an overview illustration must be created from my point of view.
Thank you for pointing this out. The Study Flow figure has been added for transparency improvement.
  • The results are presented much too briefly, furthermore only very few results are listed. If a publication should take place, these must be presented much more extensively and comprehensively.
Thank you for the comment. We have improved the results section and added a few figures to improve the transparency of the results.
  • Unfortunately, the discussion is absolutely superficial - a comprehensive revision is needed here.
Thank you for pointing this out. We have revised and rearranged the discussion section. It also should be more consistent now. Also, a few citations have been added to improve the paper's quality.

Reviewer 2 Report

This study is on the Effectiveness of an Inpatient Virtual Reality-Based Pulmonary Rehabilitation Program Among COVID-19 Patients on Symptoms of Anxiety, Depression and Quality of Life: Preliminary Results from Randomized Controlled Trial.

·         I think the manuscript is not well balanced and could be improved before publishing, especially since the results and discussion part seem too short. Please add more graphs and figures to elaborate on the results and discussion.  

·         The discussion is too short, and please categorize it into different sections and sub-sections if possible such as 4.1, 4.2, and 4.3. 4.3.1.

·         I think this paper could be a very useful study with a significant re-organization and rewriting. I suggest
re-write the abstract that presents the clear and concise message of the research paper. Avoid repetition of the words for example exercises (see lines 18-19).

·         Use any grammar-checking software to correct minor mistakes in the paper. 

·         As you can see, the discussion part is too short and lacks an obvious sub-topic. That means maybe you have not studied the data deeply. Please cite some recently published work and compare your data if possible.

·         Discuss something about genetics and the immune system. Do these factors matter during Covid-19 conditions?  

Author Response

Dear Reviewer,

Thank you for your deep and knowledgeable revision of this manuscript. The following are our answers:

  • I think the manuscript is not well balanced and could be improved before publishing, especially since the results and discussion part seem too short. Please add more graphs and figures to elaborate on the results and discussion. 

Thank you for the comment. We have improved the results section and added a few figures to improve the transparency of the results. We have revised and rearranged the discussion section. It also should be more consistent now.
  • The discussion is too short, and please categorize it into different sections and sub-sections if possible such as 4.1, 4.2, and 4.3. 4.3.1.
Thank you for pointing this out. We have revised and rearranged the discussion section. It also should be more consistent now. We tried to introduce the continuity of the narrative, so we decided not to divide the discussion into subsections
  • I suggest re-write the abstract that presents the clear and concise message of the research paper. Avoid repetition of the words for example exercises (see lines 18-19). Use any grammar-checking software to correct minor mistakes in the paper.

Thank you for the comment, changes have been made
  • As you can see, the discussion part is too short and lacks an obvious sub-topic. That means maybe you have not studied the data deeply. Please cite some recently published work and compare your data if possible.
Thank you for pointing this out. The discussion has been improved according to your comment. Also, a few citations have been added to improve the paper's quality.
  • Discuss something about genetics and the immune system. Do these factors matter during Covid-19 conditions? 

Thank you for your comment. It is certainly important to address this topic, but addressing the issue of genetics and the immune system is beyond the scope of this scientific project. The value of such a discussion would clearly increase in papers focusing on the biochemical mechanisms and epidemiology of the virus. 

Round 2

Reviewer 1 Report

The authors have implemented all of my recommendations. This improved the manuscript significantly. Congratulations for this excellent manuscript. 

Reviewer 2 Report

Acceptable with minor revisions

Thank you for giving a detailed response and the improved paper based on the previous suggestion.

Still, there is a problem with the abstract, and please don't explain the result in the abstract. Please fix it.

Please suggest,  if possible figure color can be changed. 

The conclusion is particularly poor and short; please explain more. 

Good Luck